# Diagnostic and Ethical Challenges in a Rare Case of Retroperitoneal Carcinosarcoma During Pregnancy—A Case Report and Literature Review

**DOI:** 10.3390/diagnostics15243228

**Published:** 2025-12-17

**Authors:** Marius Florentin Popa, Mihaela Lavinia Mihai, Daniela Draguta Tabirca, Mariana Deacu, Sorin Vamesu, Daniel Ioan Ureche, Vlad Iustinian Tica

**Affiliations:** 1Preclinical Sciences Department, Faculty of Medicine, Ovidius University of Constanța, 900470 Constanța, Romania; marius_popa2005@yahoo.com (M.F.P.); deacu_mariana@yahoo.com (M.D.); vtica@eeirh.org (V.I.T.); 2Forensic Medicine Department, County Clinical Emergency Hospital of Constanța, 900591 Constanța, Romania; 3Pathology Department, County Clinical Emergency Hospital of Constanța, 900591 Constanța, Romania; sorinvamesu@yahoo.com; 4Discipline of Forensic Medicine, University of Medicine and Pharmacy “Iuliu Hațieganu”, 400006 Cluj-Napoca, Romania; ureche.daniel.ioan@gmail.com; 5Romanian Academy of Scientists, 050045 Bucharest, Romania

**Keywords:** carcinosarcoma, pregnancy, retroperitoneal tumor, maternal mortality, ethical dilemmas, immunohistochemistry

## Abstract

**Background and Clinical Significance:** Carcinosarcomas are highly aggressive tumors with both carcinomatous and sarcomatous components, typically arising from the female genital tract. Primary retroperitoneal carcinosarcomas are extremely rare, and their occurrence during pregnancy presents major clinical and ethical challenges. **Case Presentation:** We report a case of a 24-year-old primigravida diagnosed with a large encapsulated retroperitoneal mass at 12 weeks of pregnancy, initially presenting with abdominal pain. The patient declined medical advice for pregnancy termination and chose to continue despite oncological risks. A multidisciplinary team planned delayed surgery after delivery. At 34 weeks, a cesarean section resulted in a healthy newborn, but surgical exploration revealed an inoperable, invasive tumor. The patient died two days later from postoperative complications. Autopsy confirmed widespread tumor invasion and lung metastases consistent with primary retroperitoneal carcinosarcoma. **Conclusions:** This case highlights the challenges of managing aggressive malignancies during pregnancy, emphasizing early diagnosis, multidisciplinary care, and ethical decision-making while respecting patient autonomy.

## 1. Introduction

Primary retroperitoneal tumors, including carcinosarcomas, are exceedingly rare, with an incidence of less than 0.2% of all malignant neoplasms [1]. These tumors arise in the retroperitoneal space from non-parenchymal structures such as adipose tissue, muscle, and connective tissues, with sarcomas being the most common malignant type. Diagnosis of these tumors is often delayed due to non-specific symptoms and is confirmed by imaging and, when possible, histopathological analysis via biopsy. Treatment primarily involves surgical resection, although the prognosis depends heavily on tumor biology and invasiveness [2].

Carcinosarcomas, or malignant mixed Mullerian tumors (MMMTs), display both carcinomatous and sarcomatous components and are highly aggressive [3]. While typically found in the female genital tract, these tumors can rarely manifest in extragenital sites, complicating diagnosis and treatment. Pregnancy further compounds the complexity of managing such tumors, presenting unique diagnostic and therapeutic challenges due to the overlapping symptoms and the need to balance maternal and fetal health [4].

We present a unique case of primary retroperitoneal carcinosarcoma in a pregnant patient, exploring the diagnostic and ethical dilemmas faced and reviewing the relevant literature on managing rare malignancies during pregnancy.

## 2. Case Report

A 24-year-old woman, 12 weeks pregnant, was admitted to our hospital with a complaint of right abdominal pain. Past medical history was unremarkable. Gynecological history included menarche at 13 years of age, regular menstrual periods, primigravida with desired pregnancy. On physical examination, blood pressure was 125/70 mmHG, heart rate 70 beats per minute (bpm), respiratory rate 16 breaths per minute, Glasgow scale 15/15, and temperature 36.7 degrees centigrade (°C); the patient had dry oral mucosa and a painful abdomen at the right middle and lower quadrant; bowel sounds were present and normal. Lower extremity edema was absent, bilaterally; singleton fetus; no vaginal discharge was observed on exploration; the gynecological exam was normal. The patient was admitted for abdominal pain. Laboratory tests showed a normal complete blood count as well as normal hepatic and renal function. Obstetric ultrasound showed biometry for 12 weeks, normally attached placenta, and normal amniotic fluid. Abdominal ultrasound revealed moderate hepatomegaly and a heterogeneous lesion measuring approximately 11 cm with a fibrous wall, findings suggestive of a possible hydatid cyst. The moderate hepatomegaly was observed only on imaging, with no clinical measurements or palpable enlargement documented on physical examination. The patient reported the resolution of abdominal pain under treatment with analgesics and antispasmodics and was discharged in good health with the recommendation to present after one month to the Surgery Clinic.

After 3 weeks she presented at hospital with pain in the right middle quadrant. Magnetic resonance imaging showing an encapsulated renal tumor 118/96/127 mm in size, non-homogeneous T2 hypersignal, with heterogeneous contrast, consistent with necrosis-type tissue. The tumor was found to exert a compressive effect, having well-defined edges with the liver, gallbladder, and the hepatic flexure of the colon. No tumor was found in other parts of the abdomen and pelvis cavity on MRI. The potential diagnoses considered included malignant renal tumor.

Given the diagnosis of retroperitoneal–renal tumor of unknown nature, a diagnostic (exploratory and bioptic) laparoscopy was requested. The kidneys appeared grossly unremarkable, with no visible tumor masses or infiltrative lesions, excluding a primary renal origin or metastatic involvement and supporting the retroperitoneal origin of the tumor. Laparoscopy was performed and revealed a voluminous tumor with diameters of approximately 13–14/10 cm; well-defined edges with the liver exerted a compressive effect displacing the diaphragmatic surface, duodenum, and hepatic flexure of the colon. Extemporaneous biopsy was performed, retrieving two cylinders from the solid component of the tumor, without complications. The histopathology report was suspicious but not conclusive for adrenal adenocarcinoma.

After that, the patient attended a multidisciplinary meeting with general surgery, gynecology, oncology, and a representative of the hospital management. The patient was, therefore, counseled for a therapeutic abortion and oncological, complex treatment. She refused as she wanted the respective baby even at the price of her life. She was supported, in that decision, by her husband and family. During the hospitalization, psychological counseling was offered to the patient, which she accepted, without changing her decision.

Following discussions within a multidisciplinary team, the patient was offered, alternatively, surgery (nephrectomy and right adrenalectomy) two weeks after delivery by caesarean section (preferably, if the patient’s health status allowed, at 34–35 weeks). The patient was discharged with a good general status and remained on monthly outpatient prenatal follow-up by maternal–fetal medicine.

Cesarean section was performed at 34 weeks, resulting in the delivery of a healthy female neonate with a birthweight of 2050 g and an Apgar score of 8. The mother’s status was not good. During the surgery, when opening the peritoneal cavity, we found blood and clots of about 400 mL. After uterine evacuation, the peritoneal cavity and retroperitoneal space were explored, and we found a retroperitoneal tumor with massive invasion, necrobiotic tissue, and superficial neoformation vessels that bled at the slightest touch.

Due to the massive invasion and associated surgical risks, the retroperitoneal tumor was not resected at that stage. The postoperative period proceeded eventfully, with complicated puerperium, and in spite of all medical efforts, the patient died 2 days later.


*The medico-legal autopsy revealed the following: *
•Metastatic tumor on the left lung [Figure 1a];•Hemoperitoneum and an impressive abdominal tumoral mass, in lysis (of 7 kg) [Figure 1c];•A tight adhesion between the tumor, liver, and adrenal gland [Figure 1d];•Gross assessment favored a tumoral origin;•A malformation on the right kidney with the renal pelvis placed at the lower pole [Figure 1b].


**Figure 1 diagnostics-15-03228-f001:**
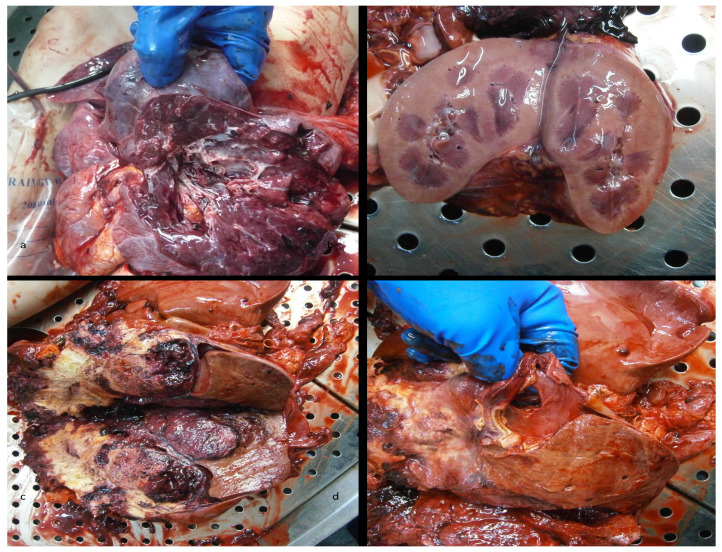
The autopsy revealed the following: (**a**) lung metastatic tumor, (**b**) malformation of the right kidney, (**c**) retroperitoneal carcinosarcoma with the invasion of the right adrenal gland, (**d**) retroperitoneal carcinosarcoma.


*Histopathology report: *
•The tumor showed biphasic differentiation.•Retroperitoneal carcinosarcoma was confirmed by immunohistochemical analysis showing Vimentin stain (+), Cytokeratin stain (+), EMA (+) [Figure 2a–c].


## 3. Discussion

Diagnosing retroperitoneal carcinosarcoma during pregnancy is inherently complex, as its symptoms—abdominal pain, distension, and discomfort—overlap with those of normal gestation. The rarity of the tumor adds to the diagnostic challenge. This case highlights the difficulty of distinguishing between pregnancy-induced symptoms and manifestations of an aggressive neoplastic process. In such scenarios, imaging plays a critical role. Magnetic resonance imaging (MRI) is particularly valuable in evaluating retroperitoneal masses, because it provides detailed soft-tissue resolution without exposing the fetus to ionizing radiation [5,6].

Similar diagnostic dilemmas have been reported in the literature. Sun et al. described a pregnant patient with a retroperitoneal Müllerian carcinosarcoma initially presenting as a cystic mass at four months’ gestation, which grew rapidly as the pregnancy progressed. This observation, consistent with our case, supports the hypothesis that elevated gestational hormones such as estrogen and progesterone may stimulate tumor growth. Such hormonal influence emphasizes the importance of high clinical suspicion and repeated imaging in pregnant women presenting with atypical or persistent abdominal symptoms [6,7].

## 4. Review of Reported Cases and Management Strategies

Retroperitoneal carcinosarcoma in pregnancy remains exceedingly rare, with only a few cases described in the literature. Most reports emphasize the tumor’s aggressive course, poor prognosis, and high metastatic potential. Booth et al. reported a retroperitoneal Müllerian carcinosarcoma associated with endometriosis, also presenting with abdominal pain and a retroperitoneal mass [7,8]. Both that case and ours underline the importance of comprehensive imaging and histopathological evaluation to distinguish benign from malignant masses during pregnancy.

Matsuo et al. (2009) reviewed sarcomas in pregnancy—mainly uterine and vulvovaginal—observing that retroperitoneal cases typically appeared later in gestation and were associated with a 5-year survival rate of only 17% [9]. Aragón-Mendoza et al. (2020) analyzed 34 case reports of retroperitoneal tumors during pregnancy, identifying that 37% were malignant, primarily sarcomas. The majority (77%) were first detected by ultrasound, and definitive surgical treatment, usually after delivery, was achieved in 88% of cases, with a maternal mortality of 8.5% [10].

Other reports describe multidisciplinary approaches adapted to gestational stage. Maglangit et al. (2021) reported a dedifferentiated liposarcoma (35 cm) treated with cesarean section at 34 weeks followed by tumor resection postpartum [11]. Similarly, Sipe et al. (2021) reported a case of retroperitoneal synovial sarcoma managed with neoadjuvant chemotherapy (doxorubicin + ifosfamide) during the second trimester, leading to tumor shrinkage and successful postpartum surgery [12].

A multicenter study by Miller et al. (2022) [13] analyzed 13 pregnant sarcoma patients treated with anthracycline/ifosfamide chemotherapy. In total, 9 of 13 pregnancies resulted in live births at an average of 31 weeks, while 4 were lost, particularly those exposed to combined regimens early in the second trimester [14]. These data underscore the need for careful therapeutic timing and tailored oncologic management to preserve both maternal and fetal outcomes [13].

## 5. Ethical and Clinical Decision-Making

This case highlights one of the most profound ethical challenges in oncology and obstetrics: managing an aggressive malignancy during pregnancy while balancing maternal and fetal well-being.

At presentation, medical recommendations favored **therapeutic**
**abortion** to enable early oncologic intervention. However, the patient made an informed and autonomous decision to continue the pregnancy despite understanding the poor maternal prognosis. This choice underscores the importance of respecting **patient autonomy**, one of the fundamental principles of medical ethics alongside **beneficence, non-maleficence, and justice** [14,15,16].

These ethical principles are also enshrined in **Romanian Law No. 46/2003 (Law on Patient Rights)**, which guarantees informed consent, the right to medical information, and the freedom to accept or refuse medical intervention.

Autonomy was upheld by ensuring that the patient was provided full disclosure of her diagnosis, prognosis, and all therapeutic options. Beneficence guided the medical team’s efforts to maximize both maternal and fetal outcomes through multidisciplinary management. Non-maleficence was respected by avoiding interventions that could cause unnecessary harm, such as high-risk chemotherapy or surgery during early pregnancy, and by postponing oncologic therapy until fetal viability. Justice was maintained by guaranteeing equitable access to multidisciplinary care, psychological support, and ethical counseling, irrespective of prognosis or personal beliefs.

## 6. Therapeutic Options Considered

The medical team discussed several therapeutic strategies with the patient:**Early surgical excision** of the retroperitoneal mass during pregnancy—rejected due to high risk of hemorrhage and fetal loss.**Neoadjuvant chemotherapy** (doxorubicin-based regimen)—contraindicated at 12 weeks due to teratogenic risk and high fetal toxicity.**Termination of pregnancy** followed by radical oncologic treatment—recommended from a curative perspective but declined by the patient.

After extensive counseling and multidisciplinary discussions involving obstetricians, oncologists, anesthesiologists, and surgeons, the patient opted for **pregnancy continuation** with planned cesarean delivery at fetal maturity, followed by surgical exploration of the tumor.

This decision represented an ethically valid exercise of autonomy, informed by complete understanding of the associated maternal risks. The medical team’s role was to respect her decision while ensuring beneficence through optimal supportive care and vigilant monitoring.

Comparable ethical dilemmas have been reported in the literature [6,7], where patients have made diverse decisions depending on gestational age, prognosis, and cultural context. In our case, despite the multidisciplinary effort, the patient’s death following cesarean delivery underscores the delicate balance between respecting patient autonomy and providing life-saving interventions in oncology during pregnancy.

## 7. Histopathological and Immunohistochemical Findings

Histopathological examination confirmed a **biphasic malignant neoplasm** composed of carcinomatous and sarcomatous elements. The tumor demonstrated spindle cell proliferation with focal epithelial differentiation and areas of necrosis, consistent with a **carcinosarcoma.**

Determining whether such an undifferentiated tumor originates from epithelial, mesenchymal, or hematopoietic lineage is essential for establishing prognosis and therapeutic direction. Immunohistochemistry (IHC) serves as a critical diagnostic tool for tumors of uncertain origin, and recent advances allow approximately 90% of morphologically ambiguous tumors to be classified using specific immunomarker panels [17].

Immunohistochemical staining revealed the following pattern:
**Marker****Expression****Interpretation**AE1/AE3PositiveConfirms epithelial component [Figure 2d]EMAPositiveSupports carcinomatous differentiationVimentinPositiveIndicates mesenchymal (sarcomatous) differentiation CK7/CK20NegativeExcludes Müllerian, pancreatobiliary, or colorectal originDesminFocallypositiveSuggests limited myogenic differentiationER/PRNegativeRules out endometrioid Müllerian originCD10NegativeAgainst endometrial stromal sarcomaS100/CD34NegativeExcludes melanocytic and vascular originRCCNegativeExcludes renal originArginase/NapsinNegativeExcludes hepatocellular and pulmonary adenocarcinoma originKi-67<10%Indicates a relatively low proliferative index

These findings confirmed a malignant biphasic tumor expressing both epithelial and mesenchymal markers, consistent with **primary retroperitoneal carcinosarcoma**—a rare but documented entity.

## 8. Exclusion of Renal Origin

Although MRI initially suggested an encapsulated “renal” tumor, the **gross and microscopic examination at autopsy revealed structurally intact kidneys** with no continuity or infiltration between the renal parenchyma and the retroperitoneal mass. The absence of macroscopic or histologic invasion effectively excludes **sarcomatoid renal cell carcinoma** or any **primary renal tumor**.

The IHC profile also supported this conclusion: **RCC marker negativity**, along with **absence of Arginase and Napsin expression**, ruled out renal, hepatic, or pulmonary adenocarcinoma. Consequently, the final diagnosis of **primary retroperitoneal carcinosarcoma** was established with high confidence [18].

The unavailability of MRI images for publication is due to institutional archiving limitations and patient privacy restrictions; however, all imaging findings are fully documented in the medical records and autopsy report.

## 9. Prognostic and Pathophysiological Considerations

This case is among the few documented instances of **maternal death due to retroperitoneal carcinosarcoma during pregnancy**, emphasizing the tumor’s aggressiveness and diagnostic difficulty. The progression coincided with gestational advancement, supporting the theory that elevated estrogen and progesterone levels may accelerate tumor proliferation [8,19].

Although **transplacental metastasis** is theoretically possible in advanced maternal malignancies, it remains exceedingly rare, primarily reported in melanoma, leukemia, and choriocarcinoma. In this case, **no placental or fetal involvement** was detected either clinically or at autopsy, and the newborn remained healthy. These findings suggest that, despite maternal tumor dissemination, the fetus was not affected by the hematogenous spread of malignant cells.

The rapid course and poor maternal outcome underscore the need for early diagnosis and standardized treatment guidelines for retroperitoneal malignancies in pregnancy. Establishing a **multidisciplinary tumor board approach** involving obstetricians, oncologists, surgeons, radiologists, and ethicists is vital to optimize both maternal and fetal outcomes [20,21,22].

## 10. Ethical Reflections and Legal Context

This case also underscores the broader ethical implications of treating malignancy during pregnancy. The patient’s decision to continue the pregnancy—fully informed and respected by the medical team—embodies the ethical principles of **autonomy, beneficence, non-maleficence, and justice**.

Under **Romanian Law No. 46/2003**, the patient’s rights to informed consent, self-determination, and refusal of medical treatment are legally protected. The healthcare team acted in accordance with these provisions, ensuring that the patient’s values and decisions were honored while striving to provide beneficent, non-maleficent care.

From an ethical perspective, this case illustrates that patient autonomy remains paramount even in life-threatening conditions. The team’s approach respected the patient’s moral and emotional convictions while maintaining professional responsibility to ensure safety, dignity, and compassionate care.

Justice was demonstrated through equal access to multidisciplinary expertise, counseling, and ongoing psychological support throughout hospitalization. The complexity of this case highlights the ethical necessity of shared decision-making, transparent communication, and documentation of informed consent in similar high-risk scenarios.

## 11. Conclusions

This case illustrates the profound ethical and medical complexities in managing rare malignancies during pregnancy, particularly when maternal and fetal health priorities may diverge. A multidisciplinary approach that integrates enhanced patient education, empathetic communication, and robust support systems is crucial to empower patients in making informed decisions. Understanding the patient’s personal motivations and concerns enables healthcare providers to address potential conflicts between clinical recommendations and the patient’s wishes, ultimately fostering trust and collaboration. This case not only contributes to the medical literature but also provides critical insights into diagnostic, therapeutic, and ethical aspects of managing retroperitoneal carcinosarcoma during pregnancy, a topic that remains poorly understood.

When maternal or fetal risk is significant, healthcare providers must respect patient autonomy while guiding them towards decisions that prioritize safety for both mother and fetus. Offering hospital resources, specialized consultations, and psychological support can help pregnant patients navigate these challenging decisions and contribute to a balanced, ethically sensitive approach.

As seen in this case, there is often a delicate balance between respecting patient autonomy and advocating for evidence-based medical care. By understanding the historical, social, and individual context of a pregnant woman’s choices, healthcare providers can better support patient-centered care that honors both maternal and fetal perspectives, particularly in life-threatening situations. Long-term prognosis appears to depend on tumor grade and completeness of resection, similar to nonpregnant cases. In summary, although evidence is limited to case reports and retrospective reviews, it consistently emphasizes that with careful, case-by-case management, women with retroperitoneal sarcoma can be treated effectively during pregnancy, optimizing outcomes for both mother and child.

## Figures and Tables

**Figure 2 diagnostics-15-03228-f002:**
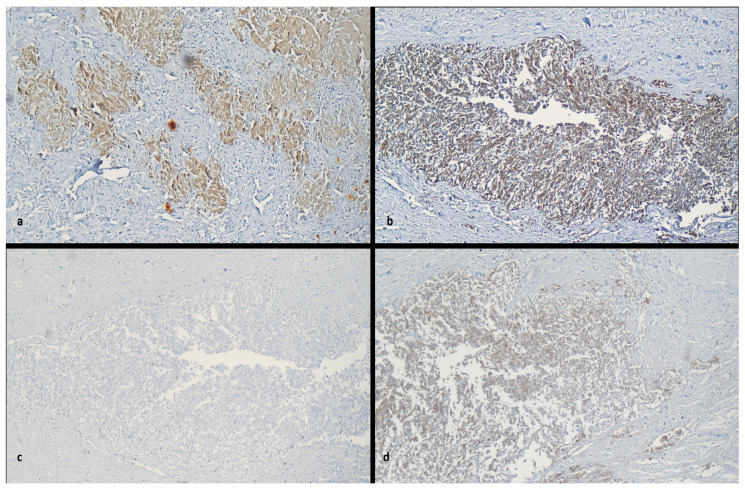
Immunohistochemical staining at x10 Ob. (**a**) Vimentin stain (+), (**b**) EMA stain (+), (**c**) CK7 stain (−), (**d**) AE1/AE3 stain (+).

## Data Availability

The data supporting this case report are derived from the patient’s medical records and are not publicly available due to privacy and ethical restrictions. Additional anonymized information may be provided by the corresponding author upon reasonable request.

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
