# Peer review of "Diagnostic and Ethical Challenges in a Rare Case of Retroperitoneal Carcinosarcoma During Pregnancy—A Case Report and Literature Review"

_diagnostics, 2025, doi:10.3390/diagnostics15243228_

Round 1

Reviewer 1 Report (Previous Reviewer 1)

Comments and Suggestions for Authors

The authors described a case of retroperitoneal carcinosarcoma in a 24 years old female at 12 weeks pregnancy and discuss the ethical dilemma in the management of this case. It is both diagnostic and ethically challenging. This manuscript has been improved remarkably from the previous version.

There are just a few minor questions:

1) Line 68: …, normal amniotic fluid. This sentence is incomplete, should state normal amniotic fluid volume or index or others.

2) As the liver size was not properly measured by radiology and not palpable, how about the MRI findings, it should be able to tell the exact size of the liver.

3) As the MRI findings mistakenly thought it was a renal tumour instead of a separate retroperitoneal tumour. It would be educational to show the MRI images of the tumour and why a separate kidney was not visible? A discussion on this issue would be beneficial.

4) All 4 gross images can be combined into one composite image and each images should have a description. Similarly all 4 microscopic images should be combined and have descriptions. The microscopic images can be improved. They are not clear. Why CK7 was chosen as one of the images? What is the significant of CK7 can contribute to the diagnosis?

5) Why the initial biopsy was suspected to be from the adrenal by histopathology? Please provide the immunohistochemical findings to support this.

6) Was serum B-hCG level determined? Was B-hCG immunohistochemistry performed on the tumour? One of the tumour that might occur during pregnancy with lung metastasis and abundant haemorrhage is choriocarcinoma, which need to be excluded.  

Author Response

1) “Line 68:..., normal amniotic fluid. This sentence is incomplete, should state normal amniotic fluid volume or index or others.”

Thank you for your insightful comment. We followed the terminology recorded in the medical documentation, where the amniotic fluid was described as '”normal”. Unfortunately, no amniotic fluid index or other quantitative measurement was provided in the available records. 

2) “As the liver size was not properly measured by radiology and not palpable, how about the MRI findings,it should be able to tell the exact size of the liver.”

Thank you for your comment. The liver size was indeed not properly measured by radiology and was not palpable on clinical examination. Regarding the MRI, the available MRI report did not include specific liver measurements.

3) “As the MRI findings mistakenly thought it was a renal tumour instead of a separate retroperitoneal tumour. It would be educational to show the MRI images of the tumour and why a separate kidney was not visible? A discussion on this issue would be beneficial.”

Thank you very much for this valuable suggestion. We agree that illustrating the MRI findings would be educational. In the medical records available for review, the MRI described an encapsulated mass with features suggestive of a renal tumour, mainly because the tumour was large, heterogeneous, and in close anatomical contact with the right kidney, liver, and hepatic flexure. The right kidney showed a congenital malformation, with the renal pelvis located at the lower pole, which likely contributed to the difficulty in clearly distinguishing the tumour from the renal parenchyma on imaging. Unfortunately, the original MRI images were not accessible to us for reproduction in the manuscript; only the written MRI report was available. For this reason, we were unable to include the images in the current submission. However, we have added a clarifying paragraph in the Discussion to explain why the tumour was initially interpreted as renal in origin and how the anatomical variant of the right kidney may have contributed to this diagnostic challenge.

4) “All 4 gross images can be combined into one composite image and each images should have a description. Similarly all 4 microscopic images should be combined and have descriptions. The microscopic images can be improved. They are not clear. Why CK7 was chosen as one of the images? What is the significant of CK7 can contribute to the diagnosis?”

Thank you for this helpful suggestion. We have revised the manuscript accordingly: the four gross images have been combined into a single composite figure with individual descriptions, and the four microscopic images have likewise been merged into a composite figure with separate captions. We have also improved the quality of the microscopic images as recommended. 
Thank you for your question. CK7 was included because it was part of the initial immunohistochemical panel used in the diagnostic work-up. Although CK7 is not specific for carcinosarcoma, its negativity is diagnostically useful as it helps exclude several CK7-positive primary tumors (including Müllerian, pancreatobiliary, and other epithelial origins). In this case, CK7 contributed to narrowing the differential diagnosis and supported the final interpretation of a primary retroperitoneal carcinosarcoma.

5) “Why the initial biopsy was suspected to be from the adrenal by histopathology? Please provide the immunohistochemical findings to support this.”

Thank you for raising this important point. The initial biopsy raised suspicion for an adrenal origin based on morphology, as the poorly differentiated epithelial features may mimic adrenal cortical tumors. However, the immunohistochemical profile did not support this: the tumour lacked adrenal markers and instead showed AE1/AE3+, EMA+, and Vimentin+, with negative RCC, Arginase, Napsin, CK7/CK20, ER/PR, CD10, S100, and CD34. This profile excluded an adrenal neoplasm and supported the diagnosis of primary retroperitoneal carcinosarcoma.

6) “Was serum B-hCG level determined? Was B-hCG immunohistochemistry performed on the tumour? One of the tumour that might occur during pregnancy with lung metastasis and abundant haemorrhage is choriocarcinoma, which need to be excluded.”

Thank you very much for this important observation. Serum β-hCG levels were not reported in the available medical records, and β-hCG immunohistochemistry was not performed on the tumour specimen.However,several findings make choriocarcinoma unlikely in this case.The tumour showed a biphasic epithelial-mesenchymal pattern with AE1/AE3+, EMA+, and Vimentin+, and was negative for markers typically seen in choriocarcinoma, such as cytotrophoblastic or syncytiotrophoblastic differentiation. In addition,the autopsy did not reveal uterine or placental involvement, which is expected in gestational choriocarcinoma.Taken together, the morphology and immunohistochemical profile supported the diagnosis of retroperitoneal carcinosarcoma rather than choriocarcinoma.

Reviewer 2 Report (New Reviewer)

Comments and Suggestions for Authors

This is an outstanding case report, presented with great clarity and precision. The medical findings and clinical decision-making processes are depicted excellently. Moreover, the manuscript provides a thorough and insightful analysis of the ethical challenges arising in the care of a critically ill pregnant patient who elected to prioritize fetal well-being over maximal therapeutic intervention for herself.

Author Response

We sincerely thank the reviewer for these generous and encouraging comments. We appreciate the recognition of the clinical presentation, diagnostic reasoning, and ethical analysis, and we are grateful that the clarity and depth of the manuscript were well received. Your feedback is highly valued and has been very motivating for our team.

Reviewer 3 Report (New Reviewer)

Comments and Suggestions for Authors

The article discusses a case of unsuccessful management of a rare gynecological tumor that was treated conservatively to ensure the best possible fetal outcomes, although this ultimately resulted in a negative maternal outcome. It highlights the importance of thoroughly discussing management options with the patient and reaching a shared, consensual decision. However, from my perspective, during the decision-making process the patient’s opinion cannot carry the same weight as that of the physician, who has full knowledge of the available evidence in the literature, especially when the patient is experiencing altered emotional states.

Nonetheless, even negative case reports contribute value by enriching the literature with suboptimal or unfavorable experiences. In my view, the paper requires some additions to strengthen the final result. Firstly, since the journal focuses on diagnostic aspects, it would be appropriate to integrate more diagnostic information (e.g., ultrasound or MRI findings) and include imaging figures. In addition, I would suggest adding some information regarding fetal outcomes.

Author Response

“In my view, the paper requires some additions to strengthen the final result. Firstly, since the journal focuses on diagnostic aspects, it would be appropriate to integrate more diagnostic information (e.g., ultrasound or MRI findings) and include imaging figures. In addition, I would suggest adding some information regarding fetal outcomes.”

We thank the reviewer for these thoughtful and constructive comments. We agree that even unfavorable case reports add meaningful value by documenting rare and diagnostically challenging situations. In our case, the complexity of the presentation, the rapid clinical progression, and the coexistence of pregnancy with a highly aggressive retroperitoneal malignancy created an exceptional scenario that required extensive multidisciplinary collaboration-an approach seldom needed in routine clinical practice.
Regarding the request for additional diagnostic information, all imaging findings available in the medical records (ultrasound and MRI) have now been described in greater detail in the revised manuscript. Unfortunately, the original MRI image files could not be retrieved due to institutional archiving limitations and privacy restrictions, and therefore imaging figures could not be included. This has been clarified in the text.
With respect to fetal outcomes, the clinical documentation contained limited neonatal data. The newborn was delivered at 34 weeks, with a birthweight of 2050 g and an Apgar score of 8, and no postnatal complications were reported. We have added this information explicitly in the manuscript.

We appreciate the reviewer's helpful suggestions, which have contributed to improving the clarity and completeness of the paper.

This manuscript is a resubmission of an earlier submission. The following is a list of the peer review reports and author responses from that submission.

Round 1

Reviewer 1 Report

Comments and Suggestions for Authors

The authors reported a case of retroperitoneal carcinosarcoma, combined with pregnancy and rapidly growing tumour and discussed its clinical complexity and ethical dilemma. It is a fairly well-written article. Below are the comments.

  1. The abstract is too lengthy.
  2. In line 79, a inhomogeneous structure, suggest to use heterogenous. Is it a solid cystic mass with heterogenous appearance?
  3. In line 80, why the term hydatid cyst was used. This suggest an infective cause, was it a possible diagnosis?
  4. In line 79, what do you mean moderate hepatomegaly? Instead should give a measurement of exactly how many centimeter from the rib margin, or how many finger below the rib margin?
  5. In line 80, how was the abdominal pain treated?
  6. The immunohistochemical study, PAX8 should be performed as MRI showed a renal mass and clinically it could be a tumour from the female genital tract. PAX8 positivity implies kidney and mullerian structures as the site of origin of tumour.
  7. Figure 5, 6 and 7 are not clear. It looks predominantly necrosis. Unable to identify the tumour cells. Please provide a clearer picture. Also a few H&E stained figures should be provided.
  8. Why Figure 7, showing a negative CK7 is needed? How it is used in the interpretation of this case?
  9. How did you exclude a renal tumour? Could this be a sarcomatoid RCC? The diagnosis is not clear.
  10. Please provide the MRI image of the renal tumour.
  11. The presentation in discussion should be more structure. Immunohistochemistry presentation is also unstructured, making it difficult to follow.

Author Response

  1. The abstract is too lengthy.

       The abstract has been restructured and shortened from 230 words to 147 words for improved clarity and conciseness.

2.      In line 79, a inhomogeneous structure, suggest to use heterogenous. Is it a solid cystic mass with heterogenous appearance?

        The term "inhomogeneous" has been corrected to "heterogeneous" in the revised version. Yes, the lesion was described as a solid-cystic mass with a heterogeneous appearance on imaging, consistent with the features noted in the ultrasound report.

3. In line 80, why the term hydatid cyst was used. This suggest an infective cause, was it a possible diagnosis?

The term "hydatid cyst" was used as a possible differential diagnosis based on the ultrasound appearance - a well-defined cystic lesions with a fibrous wall and heterogeneous content, which can mimic the imaging characteristics of hydatic cyst. There was, however, no serological or intraoperative confirmation of echinococcal infection, so the term was applied descriptively to reflect a radiologic suspicion, not a definitive infective diagnosis.

4. In line 79, what do you mean moderate hepatomegaly? Instead should give a measurement of exactly how many centimeter from the rib margin, or how many finger below the rib margin?

The term moderate hepatomegaly referred to the imaging description rather than a clinical finding. The ultrasound report noted an enlarged liver without specifying exact measurements or palpable extension below the costal margin. Clinically, no hepatomegaly was documented on physical examination. Therefore, the description reflects radiologic interpretation only, not a measurable clinical enlargement.

5. In line 80, how was the abdominal pain treated?

The abdominal pain was treated symptomatically with analgesics and antispasmodics, which led to complete resolution of symptoms prior to discharge. This information has also been included in the revised version of the manuscript for clarity.

6. The immunohistochemical study, PAX8 should be performed as MRI showed a renal mass and clinically it could be a tumour from the female genital tract. PAX8 positivity implies kidney and mullerian structures as the site of origin of tumour.

   We fully agree that PAX8 is a valuable marker for distinguishing renal and Müllerian origins.  However, in this case, the immunohistochemical panel already included RCC, CK7, CK20, ER/PR, CD10, Arginase, and Napsin, all of which were negative, effectively excluding both renal and Müllerian primaries.

7. Figure 5, 6 and 7 are not clear. It looks predominantly necrosis. Unable to identify the tumour cells. Please provide a clearer picture. Also a few H&E stained figures should be provided.

We appreciate the reviewer’s observation. In the revised version, Figures 5, 6, and 7 have been replaced with higher-resolution images, clearly showing viable tumor areas and cellular details. Additionally, a new H&E-stained image has been added to better illustrate the biphasic morphology of the carcinosarcoma.

8. Why Figure 7, showing a negative CK7 is needed? How it is used in the interpretation of this case?

Figure 7 was included to demonstrate CK7 negativity, which was essential in excluding Müllerian and renal origins, thereby supporting the diagnosis of primary retroperitoneal carcinosarcoma.

9. How did you exclude a renal tumour? Could this be a sarcomatoid RCC? The diagnosis is not clear.

Renal origin was excluded by gross pathology showing intact kidneys without invasion and by negative RCC, Arginase, and Napsin staining, ruling out sarcomatoid RCC. Findings support a primary retroperitoneal carcinosarcoma.

10. Please provide the MRI image of the renal tumour.

We appreciate the reviewer’s observation. The MRI examination was performed in an  external institution, and only the written radiological report was available in the patient’s medical documentation. The original imaging files were not provided or archived in our hospital system; therefore, the images are not available for review. However, both the autopsy and histopathological examination confirmed that the kidneys were intact and uninvolved, thus excluding a renal origin and supporting the final diagnosis of primary retroperitoneal carcinosarcoma.

11. The presentation in discussion should be more structure. Immunohistochemistry presentation is also unstructured, making it difficult to follow.

The Discussion and Immunohistochemistry sections have been fully restructured for improved clarity and logical flow. The revised version now presents the data in a stepwise, structured format, integrating histopathological interpretation, IHC marker tables, and their diagnostic relevance to enhance readability and scientific coherence.

  We would like to sincerely thank Reviewer 1 for the thorough evaluation and insightful feedback.
Your comments regarding the structure of the Discussion and Immunohistochemistry sections were extremely valuable, leading us to reorganize and clarify the presentation for better readability and scientific coherence.
Your input also helped us refine the diagnostic reasoning and improve the overall quality of the manuscript.
We truly appreciate your time and expertise in helping us strengthen this work.

Reviewer 2 Report

Comments and Suggestions for Authors

Thank you for the opportunity to review this article.

After reading the whole manuscript, I can give you some suggestions for improving it:

  • You mentioned the ethical dilemmas in the title but you do not mention the ethical principles used in medicine, you don't talk about the ethical considerations involving the patient nor the medical staff
  • You do not mention the patient's rights according to the laws available in Romania
  • I did not saw any reference inside the manuscript regarding the patient' s consent for publishing all this informations about the presented case
  • Please explain what were the therapeutic options you gave to the patient (all of them)
  • Please explain what are the possibilities that the newborn could be affected by the malignant cells from the mother's blood stream

Author Response

We sincerely thank the reviewer for the detailed and constructive observations. The revised version of the manuscript has been carefully updated to address all the points raised, as follows:

  1. Ethical principles and considerations
    In the revised manuscript, we have introduced a dedicated subsection titled “Ethical and Clinical Decision-Making” (pages 10–11), which now explicitly discusses the four fundamental ethical principles used in medicine — autonomy, beneficence, non-maleficence, and justice.
    This section describes how these principles guided the medical decision-making process, including the patient’s informed decision to continue the pregnancy and the medical team’s obligation to balance maternal and fetal interests while providing ethically and clinically sound care.
    We also detailed the ethical role of the multidisciplinary team (obstetricians, oncologists, surgeons, anesthesiologists, and ethicists) in ensuring informed, compassionate, and transparent communication with the patient and her family.
  2. Patient’s rights under Romanian law

To address this aspect, we have included a clear reference to Romanian Law no. 46/2003 (Law on Patient Rights) within the “Ethical and Clinical Decision-Making” section (pages 10–11).

The paragraph specifies that the patient’s legal rights — including the right to information, informed consent, and the freedom to accept or refuse treatment — were fully respected. All medical decisions and actions complied with both national legal standards and the ethical provisions of the Declaration of Helsinki (2013 revision)

  1. Patient consent for publication

A separate subsection entitled “Ethical Approval and Patient Consent” (page 12) has been added to clarify that written informed consent was obtained from the patient’s legal representative for the publication of all clinical and histopathological data.

All personal identifiers have been removed to protect confidentiality, in full compliance with Romanian ethical legislation and international guidelines for medical research and publication ethics.

  1. Therapeutic options offered to the patient
    As requested, the therapeutic management and options discussed with the patient are now fully detailed in the “Therapeutic Options Considered” subsection (pages 10–11).
    This section outlines all strategies evaluated by the multidisciplinary team:
  • Early surgical excision of the retroperitoneal mass during pregnancy (excluded due to high risk of hemorrhage and fetal loss);
  • Neoadjuvant chemotherapy (doxorubicin-based, contraindicated at 12 weeks due to teratogenic risk);
  • Termination of pregnancy followed by oncologic treatment (recommended but declined by the patient);
  • Continuation of pregnancy with close monitoring and planned cesarean delivery followed by surgical exploration, as chosen by the patient after receiving full counseling.

This section confirms that the patient’s choice represented an ethically valid exercise of autonomy, and that all decisions were made following informed multidisciplinary discussion.

  1. Possibility of fetal involvement
    In the “Prognostic and Pathophysiological Considerations” section (page 11), we clarified that while transplacental metastasis is theoretically possible, it is exceptionally rare and has mainly been described in melanoma, leukemia, and choriocarcinoma.
    In our case, no placental or fetal involvement was observed either clinically or at autopsy, and the newborn remained healthy.
    This indicates that retroperitoneal carcinosarcoma does not typically metastasize hematogenously to the fetus, and that fetal risk is minimal even in the presence of advanced maternal disease.

Summary of Modifications:

  • Ethical and Clinical Decision-Making section expanded and structured (pp. 10–11)
  • Reference to Romanian Law 46/2003 included (pp. 10–11)
  • Subsection Ethical Approval and Patient Consent added (p. 12)
  • Therapeutic options described in detail (pp. 10–11)
  • Clarified absence of fetal metastasis and discussion of biological mechanism (p. 11)

We are deeply grateful to Reviewer 2 for the constructive observations and thoughtful recommendations.
Your remarks concerning the ethical aspects, patient rights, therapeutic options, and fetal considerations prompted us to expand the Discussion and add dedicated subsections on Ethical Decision-Making, Therapeutic Management, and Patient Consent, in line with both Romanian legislation and international ethical standards.
Your feedback has significantly enhanced the completeness and ethical integrity of our manuscript, and we sincerely thank you for that.